# Comparative stability analysis of Indonesian banks: Markov Switching—Dynamic Regression for Islamic and conventional sectors

**Imron Mawardi[1], Muhammad Ubaidillah Al Mustofa[2], Tika Widiastuti[1], Sunan Fanani[1], Mohammed Hariri Bakri[3], Zainal Hanafi[1], Anidah Robani [3]***

**1** Department of Islamic Economics, Faculty of Economics and Business, Universitas Airlangga, Surabaya, Indonesia, **2** Department of Development Studies, Faculty of Creative Design and Digital Business, Institut Teknologi Sepuluh Nopember, Surabaya, Indonesia, **3** Institute of Technology Management and Technopreneurship, Universiti Teknikal Malaysia, Melaka, Malaysia

\* anidah@utem.edu.my

## Abstract

The banking industry necessitates implementing an early warning system to effectively identify the factors that impact bank managers and enable them to make informed decisions, thereby mitigating systemic risk. Identifying factors that influence banks in times of stability and crisis is crucial, as it ultimately contributes to developing an improved early warning system. This study undertakes a comparative analysis of the stability of Indonesian Islamic and conventional banking across distinct economic regimes—crisis and stability. We analyze monthly banking data from December 2007 to November 2022 using the Markov Switching Dynamic Regression technique. The study focuses on conducting a comparative analysis between Islamic banks, represented by Islamic Commercial Bank (ICB) and Islamic Rural Bank (IRB), and conventional banks, represented by the Conventional Commercial Bank (CCB) and Conventional Rural Bank (CRB). The findings reveal that both Islamic and conventional banks exhibit a higher probability of being in a stable regime than a crisis regime. Notably, Islamic banks demonstrate a greater propensity to remain in a stable regime than their conventional counterparts. However, in a crisis regime, the likelihood of recovery for Sharia-compliant institutions is lower than for conventional banks. Furthermore, our analysis indicates that larger banks exhibit higher stability than their smaller counterparts regarding assets and size. This study pioneers a comprehensive comparison of the Z-score, employed as a proxy for stability, between two distinct classifications of Indonesian banks: *Sharia* (ICB and IRB) and conventional (CCB and CRB). The result is expected to improve our awareness of the elements that affect the stability of Islamic and conventional banking in Indonesia, leading to a deeper comprehension of their dynamics.

**Data Availability Statement:** All relevant data are within the paper.

**Funding:** This article get supporting funding Airlangga University through the Penelitian Unggulan Airlangga (PUA) research scheme (Contract Number: 304/UN3.15/PT/2023). However, The funders played no role in preparing the study design, data collection and analysis, publication decisions, or manuscript preparation.

**Competing interests:** The authors have declared that no competing interests exist.

## Introduction

Through the provision of Islamic financial access, products and services, the Islamic Bank is one of the financial institutions that actively contributes to national economic growth. Numerous studies, including those conducted by [1–3], have demonstrated the significant contribution of Islamic banking industry to economic growth. Indonesia, which has the world's largest Muslim population of 237.55 million, or 86.7% of its total population (Royal Islamic Strategic Studies Centre), is a potential market for the growth of Islamic finance industry, particularly Islamic banking. The existence of the Islamic banking industry can be attributed to the social need for alternative financial institutions that offer reliable financial and banking services following *Sharia* rules [4]. Through a *Sharia*-compliant mechanism, Islamic banks receive deposits from surplus units and distribute them as financing to deficit units. The establishment of *Bank Muamalat* Indonesia (BMI) in 1998 directly responded to the growing demand for Islamic bank services in the market. Despite the Asian financial crisis 1998, BMI has emerged as one of the institutions that have survived and continue to operate. In 2008, the government enacted *Sharia* Commercial Bank Regulation No. 21 of 2008, to enhance the country's fundamental framework of Islamic banks.

Indonesian Islamic banking has experienced substantial growth over the past decade, with the total assets of Islamic institutions rising from 97.5 trillion Indonesian Rupiah (IDR) in 2010 to an impressive 782.1 trillion IDR in 2022. Notably, the Islamic banking industry has consistently maintained a positive return on assets in recent years. Additionally, there has been a commendable decline in Non-Performing Financing (NPF) from 4.42 percent in 2016 to 2.58 percent in 2022 for Islamic Commercial Business and from 3.42 percent in 2016 to 2.43 percent in 2022 for Sharia Business Units. These robust performances underscore the industry's effective allocation of available resources, leading to profit generation and ensuring the sustainability of their impact. Remarkably, this positive trend persisted even amid the economic recovery phase following the Covid-19-induced recession. The resilience and success of Islamic banking over the past years are indicative of its ability to navigate challenges and contribute significantly to the country's financial landscape.

As intermediary institutions, Islamic Banks grapple with various management challenges and risk exposures that pose detection challenges for Islamic Bank managers and regulators. The intricacies of these issues, coupled with insufficient supervision and a shortage of Human Resources (HR) capacity dedicated to managing Islamic banks, can lead to the emergence of deviant financial practices, ultimately causing harm to stakeholders. Such mismanagement manifests in governance inefficiencies, an erosion of public confidence in Islamic banks, rapid capital withdrawals, and poses a threat to economic stability. The potential collapse of the banking system poses a severe risk of destabilizing the economy on a massive scale, as evidenced by the devastating impacts of the 1998 banking crisis and the 2008 global financial crisis. Establishing an effective Early Warning System (EWS) is imperative to avert such catastrophes. This system should involve a meticulous analysis of factors influencing the crisis-related stability of banks. By proactively identifying and addressing these factors, Islamic banks can enhance their resilience, mitigate risks, and contribute to the overall stability of the financial system. This strategic approach can prevent the recurrence of crises, fostering a more robust and secure environment for Islamic banks and the broader economy.

In the realm of banking literature, previous studies, including those by [5–9], have contributed to the development of Early Warning System (EWS) applications. These studies have primarily focused on identifying early indicators of crises in both conventional and Sharia banking in Indonesia. The research has delved into the analysis of various factors,

encompassing both internal and external dimensions, influencing banking performance during crisis periods.

However, a critical research gap remains in comparing Z-scores between Islamic and conventional banking when constructing EWS models. Such a comparative analysis is vital as it allows for investigating government policies that can be uniformly implemented across both types of banks during a crisis. Given this gap, the present study aims to achieve several objectives. Firstly, employing the Z-Score approach, it seeks to build an early warning system model as a performance measurement tool, precisely gauging the risk of failure or bankruptcy. Secondly, employing the Markov Switching Dynamic Model approach, the study aims to investigate the determinants affecting the Z-score in two distinct regimes, namely crisis and tranquil, for two categories of Indonesian banks: Islamic (represented by Islamic Commercial Bank and Islamic Rural Bank) and conventional (represented by Conventional Commercial Bank and Conventional Rural Bank).

Furthermore, this method will facilitate the determination of the duration of each regime and the respective probabilities of their occurrence. Thirdly, the study seeks to investigate whether banks with larger assets and size exhibit a higher likelihood of operating within a stable regime than a crisis regime. The goal is to validate the argument that banks with more significant assets and size can leverage economies of scale, resulting in enhanced stability and quicker recovery from crises as studied by [10]. Through these multifaceted objectives, the study aims to contribute valuable insights to developing effective EWS models in Indonesian banking, considering both Islamic and conventional sectors. The sample of study comprises industry monthly data representing Islamic banks, specifically Islamic Commercial Banks (ICB) and Islamic Rural Banks (IRB). In parallel, conventional banks are represented by data from Conventional Commercial Banks (CCB) and Conventional Rural Banks (CRB). This selection aims to scrutinize the distinctive characteristics of these two types of banks, shedding light on their influence on the likelihood of banking vulnerability and resilience. The inclusion of these four types of banks in the sample serves the purpose of assessing overall banking stability. It is posited that both conventional and Islamic banks, particularly those with larger sizes and assets, exhibit a greater degree of stability in both crisis and tranquil regimes. The study intends to analyze certain categories to uncover insights into the distinctive traits that enhance the stability of banks, providing a full understanding of their performance in various banking situations.

The structure of this paper will be organized as follows. Chapter 2 will delve into a literature review, specifically focusing on theoretical contributions and previous literature. Chapter 3 will discuss the research methodology, highlighting data types and sources, as well as model specifications. Chapter 4 will address the findings and analysis, while Chapter 5 will explore managerial relevancies. Chapter 6 will conclude the paper, providing insights into study limitations and recommendations for further research.

## Literature review

### Dual banking system in Indonesia

Indonesia, a country with the largest Muslim population and the fourth-largest population globally, employs a dual banking system in which conventional and *Sharia*-based banks coexist and offer banking services concurrently. The fact is regulated by the Indonesian Banking Act No. 10/1998, enacted as an amendment to the Banking Act No. 7/1992. The new Banking Act states that commercial banks in Indonesia may operate conventionally (using the interest rate system) or follow *Sharia* principles. The new regulation also permits a conventional bank to establish a *Sharia*-compliant branch office. According to *Sharia* Banking Act No. 21/2008, a

*Sharia bank* can be defined as a financial institution that conducts its operations in adherence to Islamic legal principles as outlined in the fatwa issued by the *Indonesian Ulema Council (MUI)*. These principles include justice and equilibrium (*'adl wa tawazun*), promoting common good and public interest (*maslahah*), universalism (natural), and the absence of *gharar* (risk or uncertainty), *maysir* (gambling), usury, injustice, and unlawful elements. Following this, Islamic banking institutions may take the form of ICB and IRB. In contrast, conventional banking sectors could encompass CCB and CRB. This law also legalizes the spin-off of Islamic units within conventional commercial banks (*Unit Usaha Shariah*/UUS), further governed by the POJK 12 provision. This provision requires UUS with an asset value of at least fifty trillion rupiahs or fifty percent of the total value of their parent's assets to separate from their conventional bank parent.

In accordance with Indonesian Law No. 21 of 2008, the Indonesian Islamic banking sector is categorized into two distinct groups: Islamic Commercial Banks (ICB) and Islamic Rural Banks (IRB). IRBs are explicitly prohibited from engaging in demand deposit collection and providing specific payment system services. Another notable distinction is that IRBs are restricted from participating in foreign currency transactions and must be owned by an Indonesian individual or legal entity. The establishment of an ICB requires a minimum capital of IDR 3 trillion, while an IRB necessitates a comparatively lower capital. Specifically, an IRB operating within the capital city area is mandated to have a minimum capital of IDR 5 billion.

Islamic Rural Banks (IRBs) operating in the provincial capitals of Java and Bali islands must maintain a minimum capital of IDR 2 billion. For provincial capitals outside Java and Bali islands, the required minimum capital is IDR 1 billion. In other operational areas, a minimum capital of IDR 500 million is mandated. The substantial capitalization of Islamic Commercial Banks (ICB) positions them to provide a broad spectrum of financing options to both business entities and individual consumers. Conversely, IRBs primarily focus on allocating funds at a smaller financing scale and more localized level, as stipulated by POJK 66/2016 issued by the Financial Services Authority (OJK). This regulation also provides a comprehensive analysis of the similarities and differences between ICBs and IRBs. Similar distinctions apply between conventional commercial banks and rural banks.

In 2022, the Islamic banking sector comprised 13 Islamic commercial banks, 20 Islamic units or windows within conventional commercial banks (UUS), and 167 IRB. In the same year, 106 conventional commercial banks and 1,441 conventional rural banks were within the conventional banking sector. The Indonesia Financial Services Authority *(Otoritas Jasa Keuangan/OJK)* estimates that the combined assets of ICB and IRB in 2022 total IDR 552.02 trillion.

Despite the relatively low local financing market share of 7.01% (at the end of 2022) and the global market share of Islamic banking assets of 1.9% (at the end of 2021), trailing Bahrain's 3.3% and Turkey's 2.9% (the Islamic Financial Services Board, 2022), the presence of Islamic banks in Indonesia should not be overlooked. [11] assert that competition in the Indonesian Islamic bank industry significantly promotes the banking system's stability by bolstering lending activities and increasing deposit levels.

## Banking stability and crisis

According to the "too big to fail" hypothesis, the presence of larger banks makes the banking sector more vulnerable and susceptible to adverse shocks. Moreover, the prevalence of agency problems is more remarkable in larger and more diversified banks, increasing systemic risk [10]. The presence of deposit insurance and a mechanism for a lender of last resort may exacerbate this situation. Further, deposit insurance safeguards depositors against potential losses

resulting from the insolvency of a bank, thereby reducing the risk to the bank by averting a bank run and the subsequent dissemination of financial distress [12]. On the other hand, the introduction of deposit insurance could potentially undermine market discipline, leading to an increase in reckless bank conduct and moral hazard [13–16]. It has been suggested that "too big to fail" banks, which were involved in the recent global financial crisis, should be subject to more stringent regulations or be converted into smaller financial institutions [17]. In contrast, banks can leverage economies of scale through expansion, resulting in increased efficiency in intermediation, enhanced monitoring capabilities, and decreased operational costs [18]. [19] study on Ghanaian rural banking provides evidence that the size of a bank is positively correlated with its stability.

In recent research, contradictory results have emerged regarding the crisis and resilience of larger Islamic institutions, which is also the subject of an ongoing debate. A study by [10], conducted to analyze 45 Islamic banks spanning 13 countries and concluded that, relative to their lesser counterparts, larger institutions demonstrate greater stability. This contradicts the findings of [20], which investigated 76 Islamic banks in Gulf Cooperation Council (GCC) area and reached the conclusion that smaller Islamic banks are more inclined to navigate a crisis successfully. Similar findings were documented by [21], suggesting that Islamic institutions of smaller scale generally exhibit greater financial stability in comparison to those of greater scale. The observed discrepancy in results may be attributed to the increased credit risk exposure that is characteristic of larger financial institutions.

## Method

### Type and source of data

This study utilizes secondary and monthly data from ICB and IRB as representatives of the Sharia banking industry and from CCB and CRB as representatives of the conventional banking industry, spanning from December 2007 to November 2022. This study investigates various factors, encompassing the Z-Score to gauge the stability of financial institutions, as well as external and internal variables. External variables include the BI Rate and Inflation, while internal factors include Assets, Operating Ratio (OR), Available Cash (CASH), Financing to Deposit Ratio (FDR), *Mudharabah* Financing, Net Income (NI), Income Tax (ITX), Non-Performing Financing (NPF), Operating Income (OPI), Received Financing (RFIN), Return on Asset (ROA), Third Party Fund (TPF), *Ijarah* Financing, Capital Adequacy Ratio (CAR), *Murabahah* Financing, *Qard* or Loan, and Salam Financing.

The Z-Score, serving as the dependent variable ($y_t$) in this study, functions as an early warning indicator for gauging the stability and potential failure of Islamic banks. The research adopts the Z-score introduced by [21] as presented in Eq 1.

$$Z - Score = \frac{Return\ on\ Asset + Shareholder\ Equity\ Ratio}{\sigma\ Return\ on\ Asset} \qquad \text{Eq1}$$

The process of calculating the Z-score, which serves as an indicator of bank stability, involves summing the Return on Assets (ROA) and Shareholder Equity Ratio (SER). The resulting sum is then divided by the standard deviation of ROA. Notably, this approach aligns with methodologies utilized in various studies by researchers such as [7–9].

The research utilizes publicly accessible monthly and secondary data. For the BI Rate, measured as Bank Indonesia's 7-Days Repo Rate, acquired from the official website of the Central Bank of Indonesia (Bank Indonesia). The monthly inflation data is obtained from the Indonesian Statistics (BPS) using the Consumer Price Index method. The monthly inflation data is derived from the Consumer Price Index approach and obtained from the official database of

Indonesian Statistics (BPS). Data of internal variables is derived from publicly available Conventional and Islamic Banking Statistics published by the Financial Services Authority (OJK). The datasets are de-identified, without any restrictions on sharing, and are accessible to the public through their respective official websites. The research team ensures the public availability and accessibility of the data, and takes full responsibility for compliance with all applicable laws.

## Research method and model specifications

This study employs the Autoregressive Markov-Switching (MS) Model, a widely recognized approach for capturing regime transition patterns. Originating with [22, 23], seminal works in 1989 and 1990 [22, 23], analysing time series data using the MS model is instrumental in identifying characteristics within the business cycle. This econometric framework proves valuable in modelling the fluctuations of economic variables. The MS models, known for their ability to detect regime shifts, assess the duration of different regimes, and measure correlations between parameter changes in each regime, provide a robust method for understanding dynamic economic behaviour. These models aim to accommodate variations in behaviour across different states of nature, accounting for transition times between these states. The core equation of the MS model can be expressed as the following Eq 2.

$$y_t = \mu_{st} + x_t \alpha + z_t \beta_{st} + \epsilon_s \qquad \text{Eq2}$$

Where $y_t$ is the dependent variable at time $t$, $\mu_{st}$ is the state-dependent intercept (an intercept that changes with the regime $s$), $x_t$ is a vector of exogenous variables with state-invariant coefficients (coefficients that do not change with the regime) and the coefficients for these variables are denoted by $\alpha$, $z_t$ represents a vector of endogenous variables with state-dependent coefficients and the coefficients for these variables are denoted by $\beta_{st}$ and $\epsilon_s$ is independent and normally distributed errors. The error represents unobserved factors or random shocks that affect the dependent variable but are not explicitly modeled. The subscript $s$ indicates that the errors may vary across different states or regimes.

This study explains the two regimes as follows. Regime 0 is unstable and is indicative of an increased likelihood of bank failures and crises. While Regime 1 is a tranquil/stable regime, it indicates the likelihood of an increase in the bank's stability. The precision and accuracy of model predictions is contingent upon the model's specifications and using varying time parameters. Typically, the formation of a probability regime is characterized by the movement of variables. In this banking stability model, constants and variances contribute to the identification of regimes. Consequently, the model will identify a crisis if there is a substantial shift, disturbance, or significant fluctuation in the Z-score [9].

The model equation specifications are presented in Eqs 3–6. The model encompasses eight equations, as it incorporates two regimes, each comprising two equations for ICB, IRB, CCB, and CRB. The equations were simplified by excluding the lag, and lag 1 was selected based on the results of the optimum lag test.

$$ZS\ ICB = \begin{cases} \mu_0 + \beta_{01}\text{Asset} + \beta_{02}\text{BIRate} + \beta_{03}\text{OR} + \beta_{04}\text{FDR} + \beta_{05}\text{Inflation} \\ +\beta_{06}\text{Mudharabah} + \beta_{07}\text{NI} + \beta_{08}\text{NPF} + \beta_{09}\text{OPI} + \beta_{010}\text{TPF} \rightarrow \text{Regime 0} \\ \mu_1 + \beta_{11}\text{Asset} + \beta_{12}\text{BIRate} + \beta_{13}\text{OR} + \beta_{14}\text{FDR} + \beta_{15}\text{Inflation} \\ +\beta_{16}\text{Mudharabah} + \beta_{17}\text{NI} + \beta_{18}\text{NPF} + \beta_{19}\text{OPI} + \beta_{110}\text{TPF} \rightarrow \text{Regime 1} \end{cases} \qquad \text{Eq3}$$

$$ZS\ CCB = \begin{cases} \mu_0 + \beta_{01}\text{BIRate} + \beta_{02}\text{OR} + \beta_{03}\text{CASH} + \beta_{04}\text{FDR} + \beta_{05}\text{ITX} \\ +\beta_{06}\text{NPF} + \beta_{07}\text{OPI} + \beta_{08}\text{RFIN} + \beta_{09}\text{ROA} + \beta_{010}\text{TPF} \rightarrow \text{Regime 0} \\ \mu_1 + \beta_{11}\text{BIRate} + \beta_{12}\text{OR} + \beta_{13}\text{CASH} + \beta_{14}\text{FDR} + \beta_{15}\text{ITX} \\ +\beta_{16}\text{NPF} + \beta_{17}\text{OPI} + \beta_{18}\text{RFIN} + \beta_{19}\text{ROA} + \beta_{110}\text{TPF} \rightarrow \text{Regime 1} \end{cases} \quad \text{Eq4}$$

$$ZS\ IRB = \begin{cases} \mu_0 + \beta_{01}\text{Ijarah} + \beta_{02}\text{CAR} + \beta_{03}\text{Mudharabah} + \beta_{04}\text{Murabahah} + \beta_{05}\text{OPI} \\ +\beta_{06}\text{Qardh} + \beta_{07}\text{ROA} + \beta_{08}\text{Salam} + \beta_{09}\text{TPF} + \beta_{010}\text{INF} \rightarrow \text{Regime 0} \\ \mu_1 + \beta_{11}\text{Ijarah} + \beta_{12}\text{CAR} + \beta_{13}\text{Mudharabah} + \beta_{14}\text{Murabahah} + \beta_{15}\text{OPI} \\ +\beta_{16}\text{Qardh} + \beta_{17}\text{ROA} + \beta_{18}\text{Salam} + \beta_{19}\text{TPF} + \beta_{110}\text{INF} \rightarrow \text{Regime 1} \end{cases} \quad \text{Eq5}$$

$$ZS\ CRB = \begin{cases} \mu_0 + \beta_{01}\text{Asset} + \beta_{02}\text{BI Rate} + \beta_{03}\text{CAR} + \beta_{04}\text{FDR} + \beta_{05}\text{NPF} \\ +\beta_{06}\text{OPI} + \beta_{07}\text{ROA} + \beta_{08}\text{INF} \rightarrow \text{Regime 0} \\ \mu_1 + \beta_{11}\text{Asset} + \beta_{12}\text{BI Rate} + \beta_{13}\text{CAR} + \beta_{14}\text{FDR} + \beta_{15}\text{NPF} \\ +\beta_{16}\text{OPI} + \beta_{17}\text{ROA} + \beta_{18}\text{INF} \rightarrow \text{Regime 1} \end{cases} \quad \text{Eq6}$$

Finally, the implementation of the Markov Switching model was facilitated using Oxmetric Software, and the Differential Statistics Test was carried out using Stata Statistical Software.

## Results and analysis

Tables 1 illustrates the factors that influence the stability of Islamic and conventional banking in two regimes, namely Regime 0, or crisis regime, and Regime 1, or stable regime, with Table 1 focusing on the comparison between ICB and CCB and Table 2 focusing on the comparison between IRB and CRB. There are numerous intriguing variables to consider. First, the Operating Ratio (OR), derived from the ratio of Operational Costs to Operating Revenues, consistently negatively impacts the ICB and CCB Z-scores in two distinct regimes. The data indicates that elevated operational expenditures exert an adverse impact on the stability of financial institutions. The coefficient for this negative relationship was observed to be more significant for ICB during a crisis regime. This suggests that the declining stability of ICB will be influenced by high operational expenses.

Second, the Financing-to-Deposit ratio (FDR) consistently had a negative impact on the stability of the ICB and CCB, with the ICB bearing a more significant influence during crisis regimes. FDR assesses the proportion of financing to the total amount of funds and capital owned or utilized. A higher FDR suggests increased liquidity for the bank. The data suggests that banks have surplus funds, highlighting the need for effective performance of their role as financial intermediaries. During periods of instability, the increased risk of financial distress and decreased liquidity negatively affect the banking's stability [9]. The inverse correlation between FDR and banking stability suggests that as customer financing increases, so does the degree of stability exhibited by the banks. This is because banks are required to maintain adequate reserves to accommodate customer withdrawals. Over-financing is not advised, especially during periods of instability, crisis and economic failure.

Findings show that CAR is a ratio that demonstrates the banking institution's ability to provide funds to offset potential risks of loss, thereby having a positive effect on the IRB and CRB Z scores. This positive relationship was found to have the most significant coefficient on the IRB in the crisis regime, implying that adequate and available capital is required to help

**Table 1. Markov Switching model for ICB, IRB, CCB, and CRB.**

| Variable | ICB | | CCB | | IRB | | CRB | |
|---|---|---|---|---|---|---|---|---|
| | Crisis Regime—0 | Tranquil Regime—1 | Crisis Regime—0 | Tranquil Regime—1 | Crisis Regime—0 | Tranquil Regime—1 | Crisis Regime—0 | Tranquil Regime—1 |
| Constant | -0,773** | 0,119*** | 0,010** | 0,573*** | 0,234** | 0,025 | 0,056*** | 0,203*** |
| Z-score (1) | -0,07 | -0,046 | -0,047 | -0,768*** | -0,172* | -0,036 | 0,163** | -0,365*** |
| Asset | 306,759*** | -8,010*** | - | - | - | - | -9,849*** | -9,865*** |
| Asset (1) | 82,508*** | -2,028 | - | - | - | - | 2,371*** | -1,214 |
| BI Rate | 192,054* | 2,721 | -6,027*** | 290,926*** | - | - | -2,662 | 23,658* |
| BI Rate (1) | 569,694*** | 8,854 | 4,102* | -123,719*** | - | - | 4,250*** | 14,607 |
| OR | -29,906*** | -2,357** | -0,378** | -3,479*** | | | | |
| OR (1) | 50,022** | 1,079 | -0,116 | -4,147*** | | | | |
| CASH | - | - | 0,064 | -1,544** | | | | |
| CASH (1) | - | - | 0,042 | 28,377*** | | | | |
| FDR | -76,159*** | -3,203*** | -1,541*** | -31,707*** | - | - | 0,267 | -6,606*** |
| FDR (1) | 68,844*** | 4,065*** | -0,081 | -34,330*** | - | - | -0,088 | -2,444 |
| Inflation | 439,916*** | -4,196 | - | - | 0,685 | -9,941*** | 0,521 | -3,166 |
| Inflation (1) | -724,065*** | 0,791 | - | - | -2,839 | 6,433* | -1,751** | -0,225 |
| Mudharabah | -53,075*** | 1,270*** | - | - | 0,656** | 0,485 | - | - |
| Mudharabah (1) | 91,610*** | -0,958*** | - | - | -1,453*** | -0,515** | - | - |
| NI | -11,515*** | 0,135*** | - | - | | | | |
| NI (1) | 9,628*** | 0,078 | - | - | | | | |
| ITX | - | - | 0,071*** | -1,147*** | | | | |
| ITX (1) | - | - | -0,031 | -0,607*** | | | | |
| NPF | 452,035*** | -1,869 | 0,326 | -113,247*** | - | - | -3,275** | -7,653** |
| NPF (1) | 895,478*** | 14,876* | 0,341 | 41,589 | - | - | -1,637 | -6,912* |
| OPI | 19,837*** | -0,240*** | -0,096*** | 0,500*** | 0,354*** | -0,091*** | 0,019* | 0,189*** |
| OPI (1) | -19,526*** | -0,126** | 0,033** | 0,948*** | 0,025 | 0,001 | 0,003 | 0,047 |
| RFIN | - | - | -0,016 | -3,389*** | | | | |
| RFIN (1) | - | - | -0,023 | 1,593*** | | | | |
| ROA | - | - | -7,780** | -248,187*** | -91,653*** | -47,173*** | 3,057*** | -11,547 |
| ROA (1) | - | - | -0,995 | 96,712*** | 30,947 | -22,314* | 3,445*** | 12,552 |
| TPF | -310,842*** | -0,719 | -2,902*** | -15,737*** | -19,970*** | -12,036*** | - | - |
| TPF (1) | 43,831*** | 2,225 | 0,399 | -7,702*** | 11,445** | -2,592* | - | - |
| *Ijarah* | - | - | - | - | 0,630** | 0,029 | - | - |
| *Ijarah (1)* | - | - | - | - | 1,750*** | 0,126 | - | - |
| CAR | - | - | - | - | 6,966*** | 4,088*** | 0,720 | 9,066* |
| CAR (1) | - | - | - | - | -0,020 | -0,727 | 0,216 | 4,628 |
| *Murabahah* | - | - | - | - | 2,227 | 8,549*** | - | - |
| *Murabahah (1)* | - | - | - | - | -6,325* | -3,255* | - | - |
| *Qardh* | - | - | - | - | 1,547** | 0,562** | - | - |
| *Qardh (1)* | - | - | - | - | 1,829** | -0,569 | - | - |
| *Salam* | - | - | - | - | -0,110 | 0,165*** | - | - |
| *Salam (1)* | - | - | - | - | -0,440*** | 0,150*** | - | - |

Note: Significance levels are denoted by ***, **, and *, representing confidence levels of 1%, 5%, and 10%, respectively.

Source: Calculations by the Authors (2024)

**Table 2. ICB regime transition probabilities.**

| Regime Transitions | ICB | | CCB | | IRB | | CRB | |
|---|---|---|---|---|---|---|---|---|
| | Regime 0, t | Regime 1, t | Regime 0, t | Regime 1, t | Regime 0, t | Regime 1, t | Regime 0, t | Regime 1, t |
| Regime 0, t + 1 | 0,463 | 0,053 | 0,153 | 0,0688 | 0,781 | 0,087 | 0,679 | 0,133 |
| Regime 1, t + 1 | 0,537 | 0,947 | 0,847 | 0,9312 | 0,219 | 0,913 | 0,321 | 0,867 |

Source: Calculations by the Authors (2024)

stabilize the IRB in times of crisis. During a crisis, customers hoard cash instead of placing it in savings accounts or deposits. The action necessitates that banks have sufficient cash on hand, as a failure to do so could result in a bank run. Increasing CAR can increase customer security, thereby increasing customers' trust and potentially increasing bank stability. [9], who discovered the significance of CAR in influencing banking stability in both crisis and stable regimes, reached a similar conclusion. [24] emphasized the importance of capital buffers in mitigating risk in Islamic banking, where the emphasis on buffers also applies in times of crisis.

The transition matrices for regime shifts in ICB, CCB, IRB, and CRB within the Markov Switching—Dynamic Regression model are illustrated in Table 2. In this model, Regime 0 denotes the Unstable (crisis) Regime, while Regime 1 signifies the Tranquil (stable) Regime. According to Table 2, there is a 94.7% probability that ICB remains in the Non-crisis (stable) Regime when the preceding month was also in the Non-crisis Regime. Conversely, the likelihood of ICB staying in the Crisis (unstable) Regime when the prior month was also in the Crisis (unstable) Regime is 46.2%. The probability of ICB transitioning from a Stable Regime (non-crisis) to an Unstable Regime one month ago is 5.3%, while the probability of moving from a Crisis Regime (unstable) to a Stable Regime (non-crisis) one month ago is 53.7%.

Then, Table 2 depicts the regime transitions for CCB. The probability of CCB remaining in Regime 1 was 93.12% when it was also in the Tranquil Regime one month ago, while the probability of CCB remaining in Regime 2 was 15.30%. The chance for CCB to fall into the Crisis Regime (unstable) from the Stable Regime (non-crisis) was 6.88%, while the chance for ICB to move to the Stable Regime (non-crisis) from the Crisis Regime (unstable) was 68.76%.

The regime transition details for IRB are additionally displayed in Table 2. The data suggests that there is a 91.26 per cent chance that IRB will maintain its position in the Non-crisis (stable) Regime, given that it was also in the Non-crisis Regime the month prior. On the contrary, if IRB was also in the Crisis Regime the previous month, the probability of it remaining in that state is 78.08%. 8.74% is the likelihood that IRB will transition from a Stable Regime (non-crisis) to an Unstable Regime within the next month, whereas 21.92% is the likelihood that it will return to a Stable Regime (non-crisis) after a month in a Crisis Regime (unstable).

Furthermore, Table 2 offers insights into the transition probabilities or regime-switching for Conventional Rural Banks (CRB). The probability of CRB remaining in a Tranquil Regime, given its presence in the Tranquil Regime the preceding month, stands at 86.68%. Conversely, the likelihood of CRB persisting in the Crisis Regime is 67.93%. There exists a 13.32% probability of CRB transitioning from a Stable Regime (non-crisis) to a Crisis Regime when compared to its state one month prior. In contrast, the probability of CRB transitioning from a Crisis Regime to a Stable Regime (non-crisis) one month after being in a Crisis Regime is 32.07%.

Upon closer examination, it appears that Islamic banking, encompassing both ICB and IRB, is more inclined to persist within a stable framework compared to conventional banking, which includes CCB and CRB. In addition, Islamic banking has a lower likelihood of descending into a crisis regime than conventional banking. Islamic banking has a greater chance of

**Table 3. Differential statistics ICB–CCB.**

| Group | Crisis Regime—0 | | | | | | Tranquil Regime—1 | | | | | |
|---|---|---|---|---|---|---|---|---|---|---|---|---|
| | Obs | Average | Stand. Error | Stand. Deviation | 95% confidence Level | | Obs | Average | Stand. Error | Stand. Deviation | 95% confidence Level | |
| ICB | 178 | 0,091 | 0,020 | 0,271 | 0,051 | 0,131 | 178 | 0,909 | 0,020 | 0,271 | 0,869 | 0,949 |
| CCB | 178 | 0,075 | 0,017 | 0,235 | 0,041 | 0,110 | 178 | 0,925 | 0,018 | 0,235 | 0,889 | 0,959 |
| Combined | 356 | 0,083 | 0,013 | 0,253 | 0,057 | 0,109 | 356 | 0,917 | 0,013 | 0,253 | 0,890 | 0,943 |
| Diff | | 0,016 | 0,027 | | -0,037 | 0,069 | | -0,159 | 0,027 | | -0,069 | 0,037 |
| | Ha: diff < 0<br>Pr (T < t) = 0.722 | | | | | | Ha: diff < 0<br>Pr (T < t) = 0.277 | | | | | |
| | Ha: diff! = 0<br>Pr (\|T\| < \|t\|) = 0.555 | | | | | | Ha: diff! = 0<br>Pr (\|T\| < \|t\|) = 0.555 | | | | | |
| | Ha: diff > 0<br>Pr (T > t) = 0.278 | | | | | | Ha: diff > 0<br>Pr (T > t) = 0.722 | | | | | |

Source: Calculations by the Authors (20024)

remaining in a crisis regime and a lower chance of emerging from a crisis than conventional banking. The result indicates that Islamic banking is more stable and has a lower likelihood of experiencing a crisis than conventional banking. However, when Sharia banking is in a crisis, it is less able to recover (or less likely to survive) than conventional banking. Additionally, banks with more significant assets and size (ICB and CCB) have a greater chance of surviving in a stable regime and a lower chance of failing in a crisis regime than banks with smaller assets and size (IRB and CRB). Moreover, banks with greater assets and size have a greater chance of recovering from the crisis than banks with smaller sizes. The conclusion indicates that banks with larger assets and size tend to be more stable and recover from a crisis more efficiently than banks with smaller assets and size.

Table 3 –Regime 0 displays the outcome of the Differential Test for Crisis Probability between ICB and CCB. A test will be conducted using the smoothed regime probability to determine whether ICB has a lower crisis probability than CCB; the results indicate that the probability of a crisis involving ICB and CCB is statistically similar. The result means that the ICB and CCB have a similar likelihood of experiencing a crisis.

Similarly, the results of the Differential Test for the Non-Crisis Probability of ICB and CCB demonstrate the same pattern. The smoothed regime probability test determines whether ICB has a higher non-crisis probability than CCB. Table 3 –Regime 1 demonstrates the probabilities of ICB and CCB experiencing a stable regime that is not significantly different.

The results of the Differential Test for Crisis Probability between IRB and CRB are presented in Table 4 –Regime 0. A test will be conducted using the smoothed regime probability to determine whether the IRB has a lower crisis probability than the CRB. The results indicate no statistically significant difference between the two banks' crisis probability. This development implies that the IRB and CRB have nearly the same likelihood of experiencing a crisis.

Similarly, the Significant Difference Test results for IRB and CRB Non-Crisis probabilities reveal the same pattern. The smoothed regime probability will determine whether the IRB has a higher probability of non-crisis than the CRB. Table 4 –Regime 1 reveals that the IRB and CRB have non-crisis probabilities that are not statistically distinct. The result indicates that IRB and CRB have the same probability of being in a stable or tranquil regime.

The Markov Switching method allows predicting changes in the economic cycle, such as recessions, economic booms, and economic crises. This method can also be used to estimate the duration of the crisis and stable periods. The duration analysis will reveal both *Sharia* and

**Table 4. Differential statistics IRB–CRB.**

| Group | Crisis Regime—0 | | | | | | Tranquil Regime—1 | | | | | |
|---|---|---|---|---|---|---|---|---|---|---|---|---|
| | Obs | Average | Stand. Error | Stand. Deviation | 95% confidence Level | | Obs | Average | Stand. Error | Stand. Deviation | 95% confidence Level | |
| IRB | 178 | 0,293 | 0,304 | 0,406 | 0,233 | 0,353 | 178 | 0,707 | 0,304 | 0,406 | 0,647 | 0,767 |
| CRB | 178 | 0,295 | 0,029 | 0,383 | 0,239 | 0,359 | 178 | 0,705 | 0,029 | 0,383 | 0,648 | 0,761 |
| Combined | 356 | 0,294 | 0,020 | 0,394 | 0,253 | 0,335 | 356 | 0,706 | 0,021 | 0,394 | 0,665 | 0,747 |
| Diff | | -0,002 | 0,042 | | -0,085 | 0,079 | | 0,002 | 0,042 | | -0,079 | 0,085 |
| | Ha: diff < 0<br>Pr (T < t) = 0.477 | | | | | | Ha: diff < 0<br>Pr (T < t) = 0.523 | | | | | |
| | Ha: diff! = 0<br>Pr (\|T\| < \|t\|) = 0.953 | | | | | | Ha: diff! = 0<br>Pr (\|T\| < \|t\|) = 0.953 | | | | | |
| | Ha: diff > 0<br>Pr (T > t) = 0.524 | | | | | | Ha: diff > 0<br>Pr (T > t) = 0.476 | | | | | |

Source: Calculations by the Authors (2024)

conventional bank models' durations and probabilities for handling stable and crisis periods. Adjustments to research variables affect the estimation of stable and crisis periods in both models. Table 5 reveals that the ICB has 163 months of stable period with an average duration of 18.11 months and a probability of 91.57%. On the other hand, the ICB has a crisis period of 15 months with an average duration of 1.88 months and a probability of 8.43%.

Compared to the duration of Regimes 0 and 1 at CCB (Table 6), the results indicate that ICB has a longer duration of crises and a longer average duration of stability. CCB has 167 months in the stable period, with an average duration of 15.18 months and a probability of 93.82%. On the other hand, CCB has an 11-month crisis period, with an average duration of 1.1 months and a probability of 6.18 percent.

Comparable findings were observed examining the IRB and CCB, indicating that the IRB exhibited a more prolonged crisis duration and a lengthier average stable duration (Table 7). IRB has 127 months in a stable period, with an average duration of 14.11 months and a likelihood of 71.35 percent. In contrast, IRB has 51 months in the crisis period, with an average duration of 5.67 months and a probability of 28.65 percent.

**Table 5. ICB's crisis and tranquil periods.**

| ICB | | | | | |
|---|---|---|---|---|---|
| Crisis Regime—0 | | | Tranquil Regime—1 | | |
| Observed Months | Total Months | avg. prob. | Observed Months | Total Months | avg. prob. |
| 2009(11) - 2009(11) | 1 | 1 | 2008(2) - 2009(10) | 21 | 0,980 |
| 2011(3) - 2011(6) | 4 | 1 | 2009(12) - 2011(2) | 15 | 0,997 |
| 2014(10) - 2014(12) | 3 | 0,847 | 2011(7) - 2014(9) | 39 | 0,996 |
| 2017(11) - 2017(11) | 1 | 0,933 | 2015(1) - 2017(10) | 34 | 0,991 |
| 2018(5) - 2018(5) | 1 | 1 | 2017(12) - 2018(4) | 5 | 1 |
| 2020(11) - 2020(11) | 1 | 0,915 | 2018(6) - 2020(10) | 29 | 0,998 |
| 2022(1) - 2022(3) | 3 | 0,987 | 2020(12) - 2021(12) | 13 | 0,968 |
| 2022(5) - 2022(5) | 1 | 1 | 2022(4) - 2022(4) | 1 | 1 |
| - | - | - | 2022(6) - 2022(11) | 6 | 0,925 |
| Total: 15 months (8.43%) with average duration of 1.88 months. | | | Total: 163 months (91.57%) with average duration of 18.11 months. | | |

Source: Calculations by the Authors (2024)

**Table 6. CCB's crisis and tranquil periods.**

| CCB | | | | | |
|---|---|---|---|---|---|
| Crisis Regime—0 | | | Tranquil Regime—1 | | |
| Observed Months | Total Months | avg. prob. | Observed Months | Total Months | avg. prob. |
| 2008(10)-2008(10) | 1 | 1 | 2008(2)-2008(9) | 8 | 0,991 |
| 2009(4)-2009(4) | 1 | 0,993 | 2008(11)-2009(3) | 5 | 0,898 |
| 2009(9)-2009(9) | 1 | 0,944 | 2009(5)-2009(8) | 4 | 0,998 |
| 2010(12)-2010(12) | 1 | 1 | 2009(10)-2010(11) | 14 | 0,996 |
| 2011(2)-2011(2) | 1 | 1 | 2011(1)-2011(1) | 1 | 0,7 |
| 2012(1)-2012(2) | 2 | 1 | 2011(3)-2011(12) | 10 | 0,986 |
| 2015(5)-2015(5) | 1 | 0,999 | 2012(3)-2015(4) | 38 | 0,977 |
| 2017(6)-2017(6) | 1 | 1 | 2015(6)-2017(5) | 24 | 0,987 |
| 2021(9)-2021(9) | 1 | 1 | 2017(7)-2021(8) | 50 | 0,988 |
| 2022(4)-2022(4) | 1 | 0,585 | 2021(10)-2022(3) | 6 | 1 |
| - | - | - | 2022(5)-2022(11) | 7 | 1 |
| Total: 11 months (6.18%) with average duration of 1.1 months. | | | Total: 167 months (93.82%) with average duration of 15.18 months. | | |

Source: Calculations by the Authors (2024)

In contrast, CRB has 129 months in the stable period with an average duration of 9.21 months in Regime 1 and a probability of 72.47%. In contrast, CRB has a crisis period of 49 months with an average duration of Regime 0 of 3.77 months and a probability of 27.53%. Table 8 display the regime period for CRB.

Upon closer inspection, banks with fewer assets and smaller size are more likely to remain in a state of crisis for a more extended period than banks with a larger asset base and a larger size. The result confirms that banks with larger assets and size can optimize economies of scale and have more buffers than banks with smaller assets. Moreover, Islamic banks, both IRB and ICB, have a longer average duration in stable conditions than conventional banks (CCB and CRB). The outcome suggests that *Sharia* banking has the potential to be more stable than conventional banking.

**Table 7. IRB's crisis and tranquil periods.**

| IRB | | | | | |
|---|---|---|---|---|---|
| Crisis Regime—0 | | | Tranquil Regime—1 | | |
| Observed Months | Total Months | avg. prob. | Observed Months | Total Months | avg. prob. |
| 2008(2)-2009(4) | 15 | 0,978 | 2009(5)-2009(8) | 4 | 0,99 |
| 2009(9)-2009(10) | 2 | 0,789 | 2009(11)-2010(12) | 14 | 0,982 |
| 2011(1)-2011(5) | 5 | 0,931 | 2011(6)-2012(5) | 12 | 0,959 |
| 2012(6)-2012(11) | 6 | 0,777 | 2012(12)-2013(1) | 2 | 1 |
| 2013(2)-2013(6) | 5 | 0,902 | 2013(7)-2014(1) | 7 | 0,922 |
| 2014(2)-2014(3) | 2 | 0,733 | 2014(4)-2014(9) | 6 | 0,885 |
| 2014(10)-2015(4) | 7 | 0,919 | 2015(5)-2020(2) | 58 | 0,957 |
| 2020(3)-2020(4) | 2 | 0,998 | 2020(5)-2021(2) | 10 | 0,93 |
| 2021(3)-2021(9) | 7 | 0,92 | 2021(10)-2022(11) | 14 | 0,964 |
| Total: 51 months (28.65%) with average duration of 5.67 months. | | | Total: 127 months (71.35%) with average duration of 14.11 months. | | |

Source: Calculations by the Authors (2024)

**Table 8. CCB's crisis and tranquil periods.**

| CRB | | | | | |
|---|---|---|---|---|---|
| Crisis Regime—0 | | | Tranquil Regime—1 | | |
| Observed Months | Total Months | avg. prob. | Observed Months | Total Months | avg. prob. |
| 2008(8)-2008(12) | 5 | 0,944 | 2008(2)-2008(7) | 6 | 0,782 |
| 2009(8)-2010(8) | 13 | 0,923 | 2009(1)-2009(7) | 7 | 0,926 |
| 2011(6)-2011(6) | 1 | 1 | 2010(9)-2011(5) | 9 | 0,948 |
| 2012(5)-2012(5) | 1 | 0,990 | 2011(7)-2012(4) | 10 | 0,877 |
| 2013(5)-2013(7) | 3 | 0,845 | 2012(6)-2013(4) | 11 | 0,918 |
| 2013(12)-2014(9) | 10 | 0,806 | 2013(8)-2013(11) | 4 | 0,998 |
| 2015(1)-2015(1) | 1 | 0,998 | 2014(10)-2014(12) | 3 | 0,944 |
| 2015(6)-2015(7) | 2 | 0,806 | 2015(2)-2015(5) | 4 | 0,65 |
| 2017(2)-2017(6) | 5 | 0,795 | 2015(8)-2017(1) | 18 | 0,949 |
| 2019(9)-2019(11) | 3 | 0,947 | 2017(7)-2019(8) | 26 | 0,971 |
| 2020(3)-2020(3) | 1 | 0,776 | 2019(12)-2020(2) | 3 | 0,939 |
| 2021(10)-2021(11) | 2 | 1 | 2020(4)-2021(9) | 18 | 0,969 |
| 2022(2)-2022(3) | 2 | 0,888 | 2021(12)-2022(1) | 2 | 0,832 |
| - | - | - | 2022(4)-2022(11) | 8 | 0,937 |
| Total: 49 months (27.53%) with average duration of 3.77 months. | | | Total: 129 months (72.47%) with average duration of 9.21 months | | |

Source: Calculations by the Authors (2024)

## Managerial relevancies

Conventional banks, in comparison to *Sharia* banks, typically demonstrate lower stability. Unlike the conventional banking management system, which is grounded in fixed interest rates, Islamic banking operates on the principle of profit and loss sharing. The conventional banking interest system is flawed due to its susceptibility to fluctuations in interest rates, changes in macroeconomic variables, and economic cycles. Conventional banks face the risk of negative spreads, particularly when the central bank abruptly raises the benchmark interest rate. This vulnerability arises from the fact that conventional banks have disbursed loans at predetermined lending rates. The inability to promptly adjust the interest of distributed lending can lead to a negative spread, where interest income falls below the distributed financing interest and additional operational expenses. Consequently, conventional banks exhibit higher volatility and a greater reliance on market conditions, in this case, the fluctuation of central bank interest rates. In contrast, Islamic banks do not operate under the concept of negative spreads. Their revenue is derived from profit and loss sharing, aligned with real-time economic and business conditions. This distinction contributes to the enhanced stability of *Sharia* banks compared to their conventional counterparts.

Furthermore, it has been noted that Islamic banks endure longer periods of crisis in comparison to conventional banks during times of crisis. A crisis that a Sharia-compliant bank encounters results in a loss of customer confidence, which in turn extends the period required to recover. In contrast to this outcome, conventional banks encountered a crisis triggered by sudden oscillations in the reference interest rate established by the central bank, changes in macroeconomic variables, and shifts in the business cycle. Conventional banks' stability will be restored in such circumstances upon the resumption of a favorable reference interest rate and business climate.

Additionally, larger banks with greater assets and sizes tend to be more stable than those with smaller assets and sizes, according to the findings of this study. The aforementioned

results illustrate the substantial impact that size has on maintaining the stability and robustness of financial institutions. This matter necessitates deliberation due to the possibility of improved performance among Islamic banks, particularly Indonesian Islamic banks, which are facing challenges in attaining a market share surpassing 8 per cent. To optimize operational efficiency, Sharia-compliant banks must achieve a specific size threshold that allows for the realization of economies of scale. Pursuing expansion and growth is of considerable importance; however, Sharia banks must maintain their principles intact to increase their market presence and size. At a certain point, Islamic banks must exercise greater prudence regarding their susceptibility to partake in unethical practices. The greater the size of the bank, the more likely it will allocate its financing to high-risk sets and demonstrate a greater propensity for arbitrariness.

## Conclusion

Using the Markov Switching Dynamic model, this study focuses on comparing the stability-influencing factors of Islamic and conventional banking in two distinct regimes (crisis and stable regimes). In addition, this study attempts to determine the duration and probability of both crisis and stable periods. First, the results indicate that both conventional and Islamic banks have a higher chance of remaining in a stable period than in a critical one. Moreover, this probability is higher in *Sharia* banks (ICB and IRB) than in conventional banks (CCB and CB). Second, the results indicate that Islamic banks tend to remain in stable conditions for more extended periods than conventional banks, on average. Third, Islamic banks have a lower probability of entering a crisis regime than conventional banks. However, when Islamic banking enters a crisis, Islamic banks tend to remain in a crisis regime longer than conventional banks.

This research presents several limitations. Firstly, it refrains from comparing models to scrutinize variables influencing the stability of *Sharia* and conventional banks using the same set of variables. Adopting different variables in each model is intentional, aiming to derive an optimal model devoid of assumptions and potential spurious regression issues. Another area for improvement lies in the exclusive focus on ICB and IRB within the Sharia banking sample, neglecting data from the Islamic units of conventional parent banks (UUS). Future research endeavours could incorporate UUS data to achieve a more comprehensive understanding of Sharia banking. Moreover, this research is limited to the banking sector of a solitary nation, Indonesia. Further investigation is warranted to examine the stability of conventional and Sharia banking systems using samples obtained from dual banking system nations. This would enable a more comprehensive and comparative analysis.

## Author Contributions

**Conceptualization:** Imron Mawardi, Tika Widiastuti.

**Data curation:** Imron Mawardi, Muhammad Ubaidillah Al Mustofa, Sunan Fanani.

**Formal analysis:** Imron Mawardi, Muhammad Ubaidillah Al Mustofa, Tika Widiastuti, Sunan Fanani, Zainal Hanafi.

**Funding acquisition:** Imron Mawardi.

**Investigation:** Muhammad Ubaidillah Al Mustofa, Sunan Fanani, Mohammed Hariri Bakri, Zainal Hanafi.

**Methodology:** Muhammad Ubaidillah Al Mustofa, Sunan Fanani, Zainal Hanafi.

**Project administration:** Mohammed Hariri Bakri.

**Resources:** Tika Widiastuti.

**Writing – review & editing:** Anidah Robani.

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
