## [Decision Letter · Decision Letter 0]

23 Jan 2024

PONE-D-23-38458Comparative Stability Analysis of Indonesian Banks: Markov Switching - Dynamic Regression in Islamic and Conventional SectorsPLOS ONE

Dear Dr. Robani,

Thank you for submitting your manuscript to PLOS ONE. After careful consideration, we feel that it has merit but does not fully meet PLOS ONE’s publication criteria as it currently stands. Therefore, we invite you to submit a revised version of the manuscript that addresses the points raised during the review process.

We look forward to receiving your revised manuscript.

Kind regards,

Ricky Chee Jiun Chia

Academic Editor

PLOS ONE

Journal Requirements:

"This research received funding from Universitas Airlangga (UNAIR) through the Penelitian Unggulan Airlangga (PUA) research scheme (Contract Number: 304/UN3.15/PT/2023)"

Reviewers' comments:

Reviewer's Responses to Questions

**Comments to the Author**

1. Is the manuscript technically sound, and do the data support the conclusions?

Reviewer #1: Yes

Reviewer #2: Yes

2. Has the statistical analysis been performed appropriately and rigorously? 

Reviewer #1: Yes

Reviewer #2: Yes

3. Have the authors made all data underlying the findings in their manuscript fully available?

Reviewer #1: Yes

Reviewer #2: Yes

4. Is the manuscript presented in an intelligible fashion and written in standard English?

Reviewer #1: Yes

Reviewer #2: Yes

5. Review Comments to the Author

Reviewer #1: The article focuses on identifying the factors influencing the stability and crisis impact of Islamic Commercial Banks (ICB) and Islamic Rural Banks (IRB), and compares the stability of the Islamic banking industry and the traditional banking industry in Indonesia under different economic systems (crisis and stability). This is important for understanding the dynamics that affect the stability of the Islamic and traditional banking industries in the Indonesian context.

The main focus of the article is on the systemic risk of Islamic Commercial Banks (ICB) and Islamic Rural Banks (IRB) in the Indonesian context, which has a unique cultural background. However, in the fourth section "4. Results and Analysis," why is it necessary to study the impact of the Financing to Deposit Ratio (FDR) on the Industrial and Commercial Bank of China and the China Construction Bank? The banking systems of China and Indonesia are products of different economic systems, so what is the significance of this part of the research? Please provide a detailed explanation and clarification.

Reviewer #2: This is a very interesting paper describing comparative stability analysis of Indonesian Banks, using the Markov Switching -Dynamic Regression. But there are some queestions needed to revise to smooth the paper. First, the logic and format of the paper is not good, and some paragraph is repetitive. So, the author need to reorganize the paragraph and read carefully to make the logic clear and form a story, such as the table should be three-wire meter. Because the paragraph is now dissected. Second, many tables showed in the paper with different categories, so the the author should unify to one table, such as table 1 and table 2 can be merge into one table, and the decimal places oof the table should be united, which is about three decimal palces. Third, the author should make the Zt explaination of the equation 1, which is not existed in page 12.

6. PLOS authors have the option to publish the peer review history of their article (what does this mean?). If published, this will include your full peer review and any attached files.

Reviewer #1: No

Reviewer #2: No

---

## [Author Response · Author response to Decision Letter 0]

6 Mar 2024

We have removed the funding-related text from the manuscript as requested by the editorial.

Thank you for all comments. 

We have changed our manuscript according to the PLOS ONE style templates.

We would like to clarify that CCB in this manuscript is an acronym for Conventional Commercial Bank, not Commercial Bank of China or China Construction Bank. The reason we discuss FDR in this research is to determine bank liquidity conditions. Bank liquidity is important to know because it is a measure of sufficient reserves for customers to withdraw funds. We have explained this on Page 13, last paragraph.

To improve the logic clear and form a story, we have changed the draft in the INTRODUCTION and METHODOLOGY sections. Please kindly review these sections. Following the reviewer's advice, we merged several tables. The list of tables that we merge includes:

Tables 1 and 2 become Table 1

Tables 3 – 6 become Table 2

Tables 7 and 8 become Table 3

Tables 9 and 10 become Table 4.

We've also made all the decimal places of the table about three decimal places. We have also made the Zt explanation of the equation 1 based on bank type and further differentiated based on regime classification. See Equations 2 – 6.

In the "Data Availability Statement" section, we state the data that supports the findings of this study are available publicly from the Indonesian Financial Service Authority (OJK), Bank Indonesia, and Indonesian Statistics without any restriction. For more details, we provide an explanation in the Methods section on page 10, 2nd paragraph.

We also add the explanation on “Ethical Statement” section, that there is no ethical or legal restrictions on using data sets that we used in this manuscript. Kindly see page 28.

We have included the role of the funders. See Funder Role Statement on Page 28, also can be found in the cover letter.

---

## [Editor Report · Decision Letter 1]

14 Mar 2024

Comparative Stability Analysis of Indonesian Banks: Markov Switching - Dynamic Regression in Islamic and Conventional Sectors

PONE-D-23-38458R1

Dear Dr. Anidah Robani,

We’re pleased to inform you that your manuscript has been judged scientifically suitable for publication and will be formally accepted for publication once it meets all outstanding technical requirements.

Kind regards,

Ricky Chee Jiun Chia

Academic Editor

PLOS ONE
---

## [Editor Report · Acceptance letter]

24 Mar 2024

PONE-D-23-38458R1 

PLOS ONE

Dear Dr. Robani, 

I'm pleased to inform you that your manuscript has been deemed suitable for publication in PLOS ONE. Congratulations! Your manuscript is now being handed over to our production team.

Kind regards, 

on behalf of

Dr. Ricky Chee Jiun Chia 

Academic Editor

PLOS ONE